# Robust Graph Learning Against Adversarial Evasion Attacks via Prior-Free Diffusion-Based Structure Purification

## Abstract

Adversarial evasion attacks pose significant threats to graph learning, with lines of studies that have made progress in improving the robustness of Graph Neural Networks (GNNs) for real-world applications. However, existing works overly rely on priors of clean graphs or attacking strategies, which are often heuristic and not universally consistent. To achieve robust graph learning over different types of evasion attacks and diverse datasets, we investigate this non-trivial problem from a prior-free structure purification perspective. Specifically, we propose a novel **Diff**usion-based **S**tructure **P**urification framework named **DiffSP** [1], which creatively incorporates the graph diffusion model to learn intrinsic latent distributions of clean graphs and purify the perturbed structures by removing adversaries under the direction of the captured predictive patterns without relying on any pre-defined priors. DiffSP is divided into the forward diffusion process and the reverse denoising process, during which structure purification is achieved. To avoid valuable information loss during the forward process, we propose an LID-driven non-isotropic diffusion mechanism to selectively inject controllable noise anisotropically. To promote semantic alignment between the clean graph and the purified graph generated during the reverse process, we reduce the generation uncertainty by the proposed graph transfer entropy guided denoising mechanism. Extensive experiments on both graph and node classification tasks demonstrate the superior robustness of DiffSP against evasion attacks.

## Keywords

robust graph learning, adversarial evasion attack, graph structure purification, graph diffuison

## 1 Introduction

Graphs are essential for modeling relationships in web domains like social networks [69], recommendation systems [54], financial transactions [7], *etc.* While Graph Neural Networks (GNNs) [28] have advanced this field by efficiently learning representations via message passing, concerns about their robustness have arisen [26, 66, 70]. Studies show that GNNs are vulnerable to evasion adversarial attacks for in-the-wild samples [46], particularly structural perturbations [66, 71] where tiny changes to the graph topology can lead to a sharp decrease in downstream task performance. Ensuring robustness against evasion adversarial attacks is critical for the reliable application of GNNs in real-world scenarios.

A wide range of works have been proposed to enhance graph robustness, categorizing into: 1) *Structure Learning Based* methods [11, 22, 67] that focus on refining graph structures to mitigate adversarial attacks; 2) *Preprocessing Based* methods [13, 52] that focus on denoising graphs during preprocessing stage according to predefined rules; 3) *Robust Aggregation Based* methods [6, 17, 48, 70] that modify the aggregation process less sensitive to perturbations;

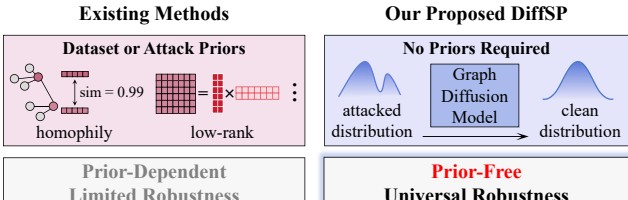

**Figure 1: Comparison of existing robust GNNs and DiffSP. Existing robust GNNs rely on priors that limit adaptability, while DiffSP is prior-free with universal robustness.**

and 4) *Adversarial Training Based* methods [58] that improve robustness by training GNNs with adversarial samples. However, most of the aforementioned approaches heavily depend on predefined priors regarding clean graphs or attack strategies [22]. For example, the homophily prior [23, 25, 64, 67] (which assumes that nodes with high feature similarity should be connected) and the low-rank prior [13, 26, 35, 57] (which assumes that the adjacency matrix of a robust graph should exhibit low-rank properties) are among the most commonly used assumptions. Unfortunately, when node features are unavailable, measuring the feature similarity becomes infeasible [22]. Additionally, imposing low-rank constraints on the graph structure risks discarding information encoded in the small singular values [11]. These prior-dependent limitations significantly hinder the ability of existing methods to achieve the universal robustness in graph learning across diverse scenarios.

To achieve prior-free robustness against adversarial evasion attacks, we aim to adaptively learn the intrinsic latent distribution from clean graphs, which captures the underlying correlation and predictive patterns to enhance the robustness of GNNs when facing unseen samples during the testing phase. Driven by this goal, we investigate this non-trivial problem from a structure purification perspective. We model the clean graph as a probability distribution over nodes and edges, encapsulating their inherent invariant and predictable properties [33]. Adversarial evasion attacks are then interpreted as disruptions to this underlying distribution, causing it to shift away from the clean distribution [30] as we observed.

To learn the latent distribution of clean graphs, the generative diffusion models [39, 49] are an ideal choice, as shown in Figure 1. Instead of relying on priors, they model the implicit distributions by fitting parameters in a data-centric manner, remaining agnostic to both the dataset and attack strategies. Unlike other generative models, the two-stage "noising-denoising" process of graph diffusion models is particularly well-suited to our goal. When encountering an attacked graph, the trained graph diffusion model gradually injects noise to obscure adversarial information during the forward diffusion process. In the reverse denoising process, step-wise denoising enables removing both the adversarial information and injected noise, achieving prior-free graph purification. Notably, the robustness gained from this framework is universally applicable, significantly enhancing its generalization on unseen test graphs.

[1]Our code is available at  https://anonymous.4open.science/r/DiffSP.

Nevertheless, it still faces two significant challenges:

1) *How can we accurately identify and remove adversarial perturbations without disrupting the unaffected portions of the graph?* Adversarial evasion attacks on graphs typically involve subtle perturbations that impact only a small subset of nodes and their associated edges, making these alterations difficult to detect [46]. During the forward diffusion process, isotropic noise is injected uniformly across the entire graph, subjecting each node to the same noise level regardless of its individual characteristics. This indiscriminate noise affects both normal and adversarial nodes, leading to excessive perturbations that can overmodify the graph. As a result, essential information may be lost, complicating the recovery of the original clean structure during the reverse denoising phase.

2) *How can we ensure that the purified graph preserves the same semantics as the target clean graph?* The generation process in diffusion models involves repeated sampling from the distribution, with the inherent randomness promoting the creation of diverse graph samples. While this diversity can be beneficial in other domains of research, it poses a significant challenge to our task of graph purification. Our objective is not to produce varied graph structures, but to accurately recover the original clean graph. Consequently, even if adversarial perturbations are successfully removed, there remains a risk that the purified graph may still diverge from the ground truth, failing to semanticly align with the target clean graph.

To address these challenges, we propose a novel **Diff**usion-based **S**tructure **P**urification framework named **DiffSP**, which creatively incorporates the diffusion model to learn the intrinsic latent distributions of clean graphs and purify the perturbed structures by removing adversaries under the direction of the captured predictive patterns without relying on any pre-defined priors. To remove adversaries while preserving the unaffected parts (▷ *Challenge 1*), we propose an LID-driven non-isotropic diffusion mechanism to selectively inject controllable noise anisotropically. By utilizing this non-isotropic noise, DiffSP effectively drowns out adversarial perturbations with minimal impact on normal nodes, thus preserving the valuable parts of the graph. To promote semantic alignment between the clean graph and the purified graph generated during the reverse process (▷ *Challenge 2*), we reduce the generation uncertainty by the proposed graph transfer entropy guided denoising mechanism. Specifically, since adversarial evasion attacks typically affect only a small portion of the graph, we maximize the transfer entropy between successive time steps during the reverse denoising process. This reduces uncertainty, stabilizes the graph generation, and guides the process toward achieving accurate graph purification. The main contributions of this paper are as follows:

- We propose DiffSP, a novel framework for adversarial graph purification against adversarial evasion attacks. To the best of our knowledge, this is the first prior-free robust graph learning framework by incorporating the graph diffusion model.
- We design an LID-driven non-isotropic forward diffusion process combined with a transfer entropy guided reverse denoising process, enabling precise removal of adversarial information while guiding the generation process toward target graph purification.
- Extensive experiments on both graph and node classification tasks on nine real-world datasets demonstrate the superior robustness of DiffSP against nine types of evasion attacks.

## 2 Related Work

**Robust Graph Learning.** Various efforts have been made to improve the robustness of graph learning against adversarial attacks, which can be grouped into four categories. 1) *Structure Learning Based* methods [11, 22, 26, 67] adjust the graph structure by removing unreliable edges or nodes to improve robustness. Pro-GNN [26] uses low-rank and smoothness regularization, GAR-NET [11] employs probabilistic models to learn a reduced-rank topology, GSR [67] leverages contrastive learning for structure refinement, and SG-GSR [22] addresses structural loss and node imbalance. 2) *Preprocessing Based* methods [13, 52] modify the graph before training. SVDGCN [13] retains top-k singular values from the adjacency matrix, while JaccardGCN [52] prunes adversarial edges based on Jaccard similarity. 3) *Robust Aggregation Based* methods [6, 17, 48, 70] improve the aggregation process to reduce sensitivity to adversarial perturbations. PA-GNN [48] and RGCN [70] use attention mechanisms to downweight adversarial edges, while Median [6] and Soft-Median [17] apply robust aggregation strategies to mitigate the effect of noisy features. 4) *Adversarial Training Based* methods [58] incorporate adversarial examples during training using min-max optimization to enhance resistance to attacks.

**Graph Diffusion Models.** Diffusion models have achieved significant success in graph generation tasks. Early works [27, 39] extended stochastic differential equations to graphs similarly to images, but faced challenges due to the discrete nature of graphs. Graph structured diffusion [18, 49] addressed this by adapting D3PM [3], improving both the quality and efficiency of graph generation. In addition, HypDiff [15] introduced a geometrically-based framework that preserves non-isotropic graph properties. To enhance scalability, EDGE [8] promotes sparsity by setting the empty graph as the target distribution. GraphMaker [31] further improved graph quality by applying asynchronous denoising to adjacency matrix and node features. However, directly applying existing graph diffusion models fails to achieve our goal because the noise injection process doesn't consider varying levels of node perturbation. This indiscriminate noise risks damaging clean nodes. Additionally, the diversity of the graph diffusion model may lead to generated graphs that fit the clean distribution but have semantic information that differs from the target clean graph.

## 3 Notations and Problem Formulation

In this work, we focus on enhancing robustness against adversarial evasion attacks with more threatening structural perturbation [71], where attackers perturb graph structures during the test phase, after the GNNs have been fully trained on clean datasets [5]. We represent a graph as $G = (\mathbf{X}, \mathbf{A})$, where $\mathbf{X}$ is the node features and $\mathbf{A}$ is the adjacency. An attacked graph is denoted as $G_{\text{adv}} = (\mathbf{X}, \mathbf{A}_{\text{adv}})$, where $\mathbf{A}_{\text{adv}}$ is the perturbed adjacency matrix. Let $c_\theta$ be the GNN classifier trained on clean graph samples, and $\mathcal{D}_{\text{test}} = \{(\hat{s}_j, y_j)\}_{j=1}^M$ represent $M$ attacked samples, where each $\hat{s}_j$ is a graph or a node, and $y_j$ is the corresponding label. The attacker's goal is to maximize the number of misclassified samples, formulated as $\max \sum_{j=1}^M \mathbb{I}(c_\theta(\hat{s}_j) \neq y_j)$, by perturbing up to $\epsilon$ edges, where $\epsilon$ is constrained by the attack budget $\Delta$. Our objective is to purify the attacked graph, reducing the effects of adversarial perturbations, and reinforcing the robustness of the GNNs to enhance the performance of downstream tasks.

**Figure 2: The overall architecture of DiffSP. DiffSP first employs a diffusion model to learn the predictive patterns of clean graphs. Then for the adversarial graph under evasion attack: 1) DiffSP injects non-isotropic noise by adjusting the diffusion time for each edge based on its adversarial degree, determined by LID. 2) During the generation process, DiffSP reduces uncertainty and guides the generation toward the target clean graph by maximizing the transfer entropy between two successive time steps.**

## 4 DiffSP

In this section, we introduce our proposed framework named DiffSP which purifies the graph structure based on the learned predictive patterns without relying on any priors about the dataset or attack strategies. The overall architecture of DiffSP is shown in Figure 2. We first present our graph diffusion purification model which serves as the backbone of DiffSP, followed by detailing the two core components: the LID-Driven Non-Isotropic Diffusion Mechanism and the Graph Transfer Entropy Guided Denoising Mechanism.

## 4.1 Graph Diffusion Purification Model

For the backbone of DiffSP, we incorporate the structured diffusion model [3, 31, 49], which has shown to better preserve graph sparsity while reducing computational complexity [18, 49]. Since we focus on the more threatening structural perturbations [71], we exclude node features from the diffusion process and keep them fixed. Specifically, the noise in the forward process is represented by a series of transition matrices, *i.e.*, $\left[\mathbf{Q}_{\mathbf{A}}^{(1)}, \mathbf{Q}_{\mathbf{A}}^{(2)}, \cdots, \mathbf{Q}_{\mathbf{A}}^{(T)}\right]$, where $(\mathbf{Q}_{\mathbf{A}}^{(t)})_{ij}$ denotes the probability of transitioning from state $i$ to state $j$ for an edge at time step $t$. The forward Markov diffusion process is defined as $q\left(\mathbf{A}^{(t)}|\mathbf{A}^{(0)}\right) = \mathbf{A}^{(0)}\mathbf{Q}_{\mathbf{A}}^{(1)} \cdots \mathbf{Q}_{\mathbf{A}}^{(t-1)} = \mathbf{A}^{(0)}\bar{\mathbf{Q}}_{\mathbf{A}}^{(t-1)}$. Here we utilize the marginal distributions of the edge state [49] as the noise prior distribution, thus $\bar{\mathbf{Q}}_{\mathbf{A}}^{(t)}$ can be expressed as $\bar{\mathbf{Q}}_{\mathbf{A}}^{(t)} = \bar{\alpha}^{(t)}\mathbf{I} + (1 - \bar{\alpha}^{(t)})\mathbf{1}\mathbf{m}_{\mathbf{A}}^{\top}$, where $\mathbf{m}_{\mathbf{A}}$ is the marginal distribution of edge states, $\bar{\alpha}^{(t)} = \cos^2\left(\frac{t/T+s}{1+s} \cdot \frac{\pi}{2}\right)$ follows the cosine schedule [38] with a small constant $s$, $\mathbf{I}$ is the identity matrix, and $\mathbf{1}$ is a vector of ones. During the reverse denoising process, we use the transformer convolution layer [42] as the denoising network $\phi(\cdot)_{\boldsymbol{\theta}}$, trained for one-step denoising $p_{\boldsymbol{\theta}}\left(\mathbf{A}^{(t-1)}|\mathbf{A}^{(t)}, t\right)$. We can train the denoising network to predict $\mathbf{A}^{(0)}$ instead of $\mathbf{A}^{(t-1)}$ since the posterior $q\left(\mathbf{A}^{(t-1)}|\mathbf{A}^{(t)}, \mathbf{A}^{(0)}, t\right) \propto \mathbf{A}^{(t)}(\mathbf{Q}_{\mathbf{A}}^{(t)})^{\top} \odot \mathbf{A}^{(0)}\bar{\mathbf{Q}}_{\mathbf{A}}^{(t-1)}$ has a closed form expression [31, 44, 45], where $\odot$ is the Hadamard product. Once trained, we can generate graphs by iteratively applying $\phi(\cdot)_{\boldsymbol{\theta}}$.

## 4.2 LID-Driven Non-Isotropic Diffusion Mechanism

Adversarial attacks typically target only a small subset of nodes or edges to fool the GNNs while remaining undetected. Injecting isotropic noise uniformly across all nodes, which means applying the same level of noise to each node regardless of its individual characteristics [50], poses a significant challenge. While isotropic noise can effectively drown out adversarial perturbations during the forward diffusion process, it inevitably compromises the clean and unaffected portions of the graph. As a result, both the adversarial and the valuable information are erased, making purification during the reverse denoising process more difficult.

To remove the adversarial perturbations without losing valuable information, we design a novel LID-Driven Non-Isotropic Diffusion Mechanism. The core idea is to inject more noise into adversarial nodes identified by Local Intrinsic Dimensionality (LID) while minimizing disruption to clean nodes. In practice, the noise level associated with different edges is distinct and independent. As a result, the noise associated with each edge during the forward diffusion process is represented by an independent transition matrix. The adjacency matrix $\mathbf{A}^{(t)}$ at time step $t$ is then updated as follows:

$$\mathbf{A}_{ij}^{(t)} = \mathbf{A}_{ij}(\bar{\mathbf{Q}}_{\mathbf{A}}^{(t)})_{ij}, \tag{1}$$

$$(\bar{\mathbf{Q}}_{\mathbf{A}}^{(t)})_{ij} = \bar{\alpha}^{(t)}\mathbf{I} + (\mathbf{\Lambda}_{\mathbf{A}})_{ij}(1 - \bar{\alpha}^{(t)})\mathbf{1}\mathbf{m}_{\mathbf{A}}^{\top}, \tag{2}$$

where $\mathbf{\Lambda}_{\mathbf{A}} \in \mathbb{R}^{N \times N}$ represents the adversarial degree of each edge.

Based on the above analysis, locating the adversarial information and determining the value of $\mathbf{\Lambda}_{\mathbf{A}}$ is crucial for effective adversarial purification. Local Intrinsic Dimensionality (LID) [21, 37] measures the complexity of data distributions around a reference point $o$ by assessing how quickly the number of data points increases as the distance from the reference point expands. Let $F(r)$ denote the cumulative distribution function of the distances between the reference point $o$ and other data points at distance $r$ and $F(r)$ is

positive and differentiable at $r \geq 0$, the LID of point $o$ at distance $r$ is defined as $\lim_{\epsilon \to 0} \frac{\ln F((1+\epsilon)r)/F(r)}{\ln(1+\epsilon)}$ [21]. According to the manifold hypothesis [14], each node $n_i$ in a graph lies on a low-dimensional natural manifold $S$. Adversarial nodes being perturbed will deviate from this natural data manifold $S$, leading to an increase in LID [37], which can quantify the dimensionality of the local data manifold. Therefore, we use LID to measure the adversarial degree, $\Lambda_A$. Higher LID values indicate that the local manifold around a node has expanded beyond its natural low-dimensional manifold $S$, signaling the presence of adversarial perturbations. In this work, we use the Maximum Likelihood Estimator (MLE) [2] to estimate the LID value of graph nodes, providing a useful trade-off between statistical efficiency and computational complexity [37]. Specifically, let $\Gamma \in \mathbb{R}^n$ represent the vector of estimated LID values, where $\Gamma_i$ denotes the LID value of node $n_i$, which is estimated as follows:

$$\Gamma_i = -\left(\frac{1}{k}\sum_{j=1}^{k}\log\frac{r_j(n_i)}{r_k(n_i)}\right)^{-1}. \tag{3}$$

Here, $r_j(n_i)$ represents the distance between node $n_i$ and its $j$-th nearest neighbor $n_i^j$. Based on the observation that the deeper layers of a neural network reveal more linear and "unwrapped" manifolds compared to the input space [16], we compute the $r_j(n_i)$ as the Euclidean distance [12] between the hidden features of two nodes in the last hidden layer of the trained GNN classifier $c(\cdot)_\theta$. After obtaining the LID values vector $\Gamma$, we can calculate $\Lambda_A = \Gamma\Gamma^\top$.

However, in practice, using the non-isotropic transition matrix in Eq. (1) requires the diffusion model to predict the previously injected non-isotropic noise during the reverse process. This task is more challenging because, unlike isotropic noise, non-isotropic noise varies across different edges. As a result, the model must learn to predict various noise distributions that are both spatially and contextually dependent on the graph structure and node features. This increases the difficulty of accurately estimating and removing the noise across graph regions, making the reverse denoising process significantly more intricate. Moreover, training the model to develop the ability to inject more noise into adversarial perturbations and remove it during the reverse process relies on having access to adversarial training data. However, in the evasion attack settings, where the model lacks access to adversarial graphs during training, its ability to achieve precise non-isotropic denoising is limited. Inspired by [63], we introduce the following proposition:

PROPOSITION 1. *For each edge at time $t$, the adjacency matrix is updated as $A_{ij}^{(t)} = A_{ij}(\bar{Q}_A'^{(t)})_{ij}$, where the non-isotropic transition matrix is $(\bar{Q}_A'^{(t)})_{ij} = \bar{\alpha}^{(t)}I + (\Lambda_A)_{ij}(1-\bar{\alpha})\mathbf{1}\mathbf{m}_A^T$. There exists a unique time $\hat{t}(A_{ij}) \in [0, T]$ such that $(\bar{Q}_A'^{(t)})_{ij} \Leftrightarrow (\bar{Q}_A^{\hat{t}(A_{ij})})_{ij}$, where:*

$$\hat{t}(A_{ij}) = T\left(\frac{2(1+s)}{\pi}\cos^{-1}\left(\sqrt{\frac{\bar{\alpha}^{(t)}}{[\Lambda(A)_{ij}(1-\bar{\alpha}^{(t)})+\bar{\alpha}^{(t)}]}}\right)-s\right). \tag{4}$$

This proposition demonstrates that non-isotropic noise can be mapped to isotropic noise by adjusting the diffusion times accordingly. The detailed proof is provided in Appendix A.1. Building on this proposition, we bypass the need to train a diffusion model that can predict non-isotropic noise in the reverse denoising process. Instead, we handle the need for non-isotropic noise injection by

applying isotropic noise uniformly to all edges, while varying the total diffusion time for each edge. By controlling the diffusion time for each edge, we can effectively manage the noise introduced to each node, ensuring that the injected noise accounts for the adversarial degree of each node. Let $\hat{A}^{(t)\prime}$ represents the adjacency matrix at time $t$ during the reverse denoising process, we have:

$$\hat{A}^{(t)\prime} = M^{(t)} \odot \hat{A}^{(t)} + (1 - M^{(t)}) \odot A^{(t)}, \tag{5}$$

where $\hat{A}^{(t)}$ is the adjacency matrix predicted by $\phi(\cdot)_\theta$, $A^{(t)}$ is the noisy adjacency matrix obtained by $A^{(t)} = A\bar{Q}_A^{(t)}$ in the forward diffusion process, and $M^{(t)}$ is the binary mask matrix that indicates which edges are being activated to undergo purification at time step $t$, achieving the non-isotropic diffusion. $M_{ij}^{(t)}$ is defined as:

$$M_{ij}^{(t)} = \begin{cases} 0, & t > \hat{t}(A_{ij}) \\ 1, & t \leq \hat{t}(A_{ij}) \end{cases}, \tag{6}$$

where $\hat{t}(A_{ij})$ is obtained according to Proposition 1. This implies that clean nodes are not denoised until the specified time. In this way, adversarial information receives sufficient denoising, while valuable information is not subjected to excessive perturbations.

## 4.3 Graph Transfer Entropy Guided Denoising Mechanism

In structured diffusion models [3], the reverse process involves multiple rounds of sampling from the distribution, which introduces inherent randomness. This randomness is useful for generating diverse graph samples but creates challenges for our purification goal. During the reverse denoising process, the diversity of diffusion can result in purified graphs that, although free from adversarial attacks and fit the clean distribution, deviate from the target graph and have different ground truth labels. This presents a significant challenge: we not only encourage the generated graph to be free from adversarial information but also aim for it to retain the same semantic information as the target clean graph.

To address this challenge, we introduce a Graph Transfer Entropy Guided Denoising Mechanism to minimize the generation uncertainty in the reverse Markov chain $\langle \hat{G}^{(T-1)} \to \hat{G}^{(T-2)} \to \cdots \to \hat{G}^{(0)} \rangle$. Transfer entropy [40] is a non-parametric statistic that quantifies the directed transfer of information between random variables. The transfer entropy from $\hat{G}^{(t)}$ to $\hat{G}^{(t-1)}$ in the reverse process by knowing the adversarial graph $G_{\text{adv}}$, can be defined in the form of conditional mutual information [56]:

$$I(\hat{G}^{t-1}; G_{\text{adv}}|\hat{G}_t) = H(\hat{G}^{(t-1)}|\hat{G}^{(t)}) - H(\hat{G}^{(t-1)}|\hat{G}^{(t)}, G_{\text{adv}}), \tag{7}$$

where $I(\cdot)$ represents mutual information and $H(\cdot)$ is the Shannon entropy. This measures the uncertainty reduced about future value $\hat{G}^{(t-1)}$ conditioned on the value $G_{\text{adv}}$, given the knowledge of past values $\hat{G}^{(t)}$. Given the unnoticeable characteristic of adversarial attacks, which typically involve only small perturbations to critical edges without altering the overall semantic information of most nodes, the target clean graph has only minimal differences from $G_{\text{adv}}$. Therefore, by increasing the $I(\hat{G}^{t-1}; G_{\text{adv}}|\hat{G}_t)$, we can mitigate the negative impacts of generative diversity on our goal and guide the direction of the denoising process, ensuring that the generation towards the target clean graph. Specifically, the purified graph will not only be free from adversarial attacks but will also share the same semantic information as the target clean graph.

However, calculating Eq. (7) requires estimating both the entropy and joint entropy of graph data, which remains an open problem.

In this work, we propose a novel method for estimating graph entropy and joint entropy. Let $z_i$ be the representations of node $n_i$ after message passing. By treating the set $\mathcal{Z} = \{z_1, z_2, \ldots, z_n\}$ as a collection of variables that capture both feature and structure information of the graph, we approximate it as containing the essential information of the graph. From this perspective, the entropy of the graph can be estimated using matrix-based Rényi $\alpha$-order entropy [62], which provides an insightful approach to calculating the graph entropy. Specifically, let $\mathbf{K}$ denote the Gram matrix obtained from evaluating a positive definite kernel $k$ on all pairs of $z$ with $\mathbf{K}_{ij} = \exp\left(-\frac{\|z_i - z_j\|^2}{2\sigma^2}\right)$, where $\sigma$ is a hyperparameter selected follows the Silverman's rule [43], the graph entropy can then be defined as the Rényi's $\alpha$-order entropy $S_\alpha(\cdot)$ [62]:

$$H(G) = S_\alpha(\hat{\mathbf{K}}) = \frac{1}{1-\alpha} \log \left[ \sum_1^n \lambda_i^\alpha(\hat{\mathbf{K}}) \right], \qquad (8)$$

where $\hat{\mathbf{K}}_{ij} = \frac{1}{n} \frac{\mathbf{K}_{ij}}{\sqrt{\mathbf{K}_{ii} \mathbf{K}_{jj}}}$, $\lambda_i(\hat{\mathbf{K}})$ denotes the $i$-th eigenvalue of $\hat{\mathbf{K}}$, and $\alpha$ is a task-dependent parameter [62]. In the context of graph learning, Eq. (8) captures the characteristics of the graph's community structure: lower graph entropy signifies a more cohesive and well-defined community structure, whereas higher graph entropy indicates a more disordered and irregular arrangement. Further details can be found in Appendix B. For a collection of $m$ graphs with their node representations after message passing $\{\mathcal{Z}_i = (z_1^i, z_2^i, \cdots, z_n^i)\}_{i=1}^m$, the joint graph entropy is defined as [62]:

$$H(G_1, G_2, \cdots, G_m) = S_\alpha \left( \frac{\hat{\mathbf{K}}_1 \odot \hat{\mathbf{K}}_2 \odot \cdots \odot \hat{\mathbf{K}}_m}{\mathrm{tr}\left(\hat{\mathbf{K}}_1 \odot \hat{\mathbf{K}}_2 \odot \cdots \odot \hat{\mathbf{K}}_m\right)} \right), \qquad (9)$$

where $\hat{\mathbf{K}}_i$ is the normalized Gram matrix of $G_i$, $\odot$ represents the Hadamard product, and $\mathrm{tr}(\cdot)$ is the matrix trace. Further understanding of our calculation method can be found in Appendix B.

By combining Eq. (8) and Eq. (9), we can get the value of transfer entropy $I(\hat{G}^{(t-1)}; G_{\mathrm{adv}} | \hat{G}^{(t)})$. The detailed derivation process is provided in Appendix A.2. Intuitively, based on our entropy estimation method, maximizing $I(\hat{G}^{(t-1)}; G_{\mathrm{adv}} | \hat{G}^{(t)})$ will guide the node entanglement of the generated $\hat{G}^{(t-1)}$ towards that of $G_{\mathrm{adv}}$, preventing the reverse denoising process from deviating from the target direction. To achieve this, we update the generation process using the negative gradient of $I(\hat{G}^{(t-1)}; G_{\mathrm{adv}} | \hat{G}^{(t)})$ concerning $\hat{\mathbf{A}}^{(t-1)}$:

$$\hat{\mathbf{A}}^{(t-1)} \leftarrow \hat{\mathbf{A}}^{(t-1)} + \lambda \nabla_{\hat{\mathbf{A}}^{(t-1)}} I(\hat{G}^{(t-1)}; G_{\mathrm{adv}} | \hat{G}^{(t)}), \qquad (10)$$

where $\lambda$ is a hyperparameter controlling the guidance scale. Early in the denoising process, maximizing the $I(\hat{G}^{(t-1)}; G_{\mathrm{adv}} | \hat{G}^{(t)})$ will steer the overall direction of the generation toward better purification. However, as the graph becomes progressively cleaner, maintaining the same level of guidance could cause the re-emergence of adversarial information in the generated graph. Therefore, it is essential to adjust the guidance scale dynamically over time. We propose that the scale of guidance should depend on the ratio between the injected noise and the adversarial perturbation at each time step. We update the guidance process in Eq. (10) as follows:

$$\hat{\mathbf{A}}^{(t-1)} \leftarrow \hat{\mathbf{A}}^{(t-1)} - \frac{\lambda}{1 - \bar{\alpha}} \nabla_{\hat{\mathbf{A}}^{(t-1)}} I(\hat{G}^{(t-1)}; G_{\mathrm{adv}} | \hat{G}^{(t)}). \qquad (11)$$

## 4.4 Training Pipeline of DiffSP

Under evasion attacks, we train the proposed DiffSP and the classifier with the overall objective $\mathcal{L} = \mathcal{L}_{\mathrm{cls}} + \mathcal{L}_{\mathrm{diff}}$, where:

$$\mathcal{L}_{\mathrm{cls}} = \mathrm{cross\text{-}entropy}(\hat{y}, y), \qquad (12)$$

$$\mathcal{L}_{\mathrm{diff}} = \mathbb{E}_{q(\mathbf{A}^{(0)})} \mathbb{E}_{q(\mathbf{A}^t | \mathbf{A}^{(0)})} \left[ -\log p_\theta(\mathbf{A}^{(0)} | \mathbf{A}^{(t)}, t) \right]. \qquad (13)$$

The classifier loss $\mathcal{L}_{\mathrm{cls}}$ measures the difference between the predicted label $\hat{y}$ and the ground truth $y$. The graph diffusion model loss $\mathcal{L}_{\mathrm{diff}}$ accounts for the reverse denoising process [3]. Initially, we train the classifier, followed by the independent training of the diffusion model. Once both models are trained, they are used together to purify adversarial graphs. The training pipeline of DiffSP is detailed in Algorithm 1, and complexity analysis is in Appendix C.

---

**Algorithm 1:** Overall training pipeline of **DiffSP**.

---

**Input:** Evasion attacked graph $G_{\mathrm{adv}} = (\mathbf{X}, \mathbf{A}_{\mathrm{adv}})$; Classifier $c(\cdot)_\theta$; Graph diffusion purification model $\phi(\cdot)_\theta$; Hyperparameters $\hat{T}, k, \lambda, \sigma, \alpha, \eta$.

**Output:** Purified graph $\hat{G} = (\mathbf{X}, \hat{\mathbf{A}})$; Learned parameter $\hat{\theta}$.

1 Update by back-propagation $\hat{\theta} \leftarrow \hat{\theta} - \eta \nabla_{\hat{\theta}} \mathcal{L}$ ;

   // LID-Driven Non-Isotropic Diffusion

2 Assess node adversarial degree $\Gamma$ based on LID $\leftarrow$ Eq. (3);

3 Calculate the edge adversarial degree $\Lambda_{\mathbf{A}} = \Gamma\Gamma^\top$;

4 Obtain the purification time of each edge $\hat{t}(\mathbf{A}_{ij}) \leftarrow$ Eq. (4);

5 **for** $t = T, T-1, \cdots, 1$ **do**

6     Establish the purification mask $\mathbf{M}^{(t-1)} \leftarrow$ Eq. (6);

7     Execute one step denoising $\hat{\mathbf{A}}^{(t-1)} \leftarrow$ Eq. (5);

      // Graph Transfer Entropy Guided Denoising

8     Calculate the graph transfer entropy $\leftarrow$ Eq. (7), (8), (9);

9     Guide the reverse denoising process $\leftarrow$ Eq. (11);

10 Obtain the $\hat{\mathbf{A}}^{(0)}$ as the purified adjacency matrix $\hat{\mathbf{A}}$.

---

## 5 Experiment

In this section, we conduct extensive experiments on graph and node classification tasks to evaluate the robustness of DiffSP against various adversarial evasion attacks. We first introduce the experiment settings and then present the results.

### 5.1 Experiment Settings

**Datasets.** We assess the robustness of DiffSP in both graph classification and node classification tasks. For graph classification, we use MUTAG [24], IMDB-BINARY [24], IMDB-MULTI [24], REDDIT-BINARY [24], and COLLAB [24] datasets. For node classification, we test on Cora [59], CiteSeer [59], Polblogs [1], and Photo [41]. We apply an 8:1:1 random split for graph classification and a 1:1:8 random split for node classification. Details are in Appendix D.1.

**Baselines.** Due to the limited research on robust GNNs targeting graph classification under adversarial attacks, we compare DiffSP with robust representation learning and structure learning methods designed for graph classification, including IDGL [9], GraphCL [60], VIB-GSL [47], G-Mixup [20], SEP [53], MGRL [36], SCGCN [68], HSP-SL [65], SubGattPool [4] DIR [55], and VGIB [61]. For node classification, we choose baselines from four categories:

**Table 1: Accuracy score (% ± standard deviation) of *graph classification* task on real-world datasets against adversarial attacks. The best results are shown in bold type and the runner-ups are underlined. OOM indicates out-of-memory.**

| Dataset | Attack | GCN | IDGL | GraphCL | VIB-GSL | G-Mixup | SEP | MGRL | SCGCN | HSP-SL | SubGattPool | DIR | VGIB | DiffSP |
|---|---|---|---|---|---|---|---|---|---|---|---|---|---|---|
| **MUTAG** | GradArgmax | 54.44±4.16 | 47.78±5.09 | 55.00±3.89 | _69.45±2.78_ | 60.00±5.44 | 62.78±4.34 | 66.67±3.51 | 67.22±3.89 | 68.33±3.51 | 62.78±3.56 | 54.44±4.16 | 68.52±2.62 | **70.00±4.44** |
| | PR-BCD | 51.66±5.00 | 65.00±6.11 | 59.44±6.11 | 62.77±4.34 | _71.11±3.33_ | 55.00±9.44 | 56.67±4.16 | 63.89±2.78 | 65.56±2.22 | 65.56±5.98 | 52.77±6.21 | 57.78±3.68 | **72.77±6.31** |
| | CAMA | 40.56±2.54 | **73.89±2.55** | 44.26±3.80 | 59.34±3.52 | 60.00±2.22 | 60.56±1.67 | 39.45±1.67 | 64.45±7.11 | 43.33±2.22 | 66.11±5.80 | 62.22±8.17 | 61.67±1.67 | 68.33±9.31 |
| | Average | 48.89 | 62.22 | 52.90 | 64.04 | 63.70 | 59.45 | 54.26 | 65.19 | 59.07 | _64.82_ | 56.48 | 62.66 | **70.18** |
| **IMDB-B** | GradArgmax | 62.79±1.08 | 59.20±1.08 | 65.19±0.87 | 68.90±1.45 | 50.89±0.20 | _72.00±1.55_ | 64.00±0.77 | 68.60±1.50 | 62.50±0.80 | 61.00±1.10 | 68.10±1.04 | 63.80±0.87 | **76.00±1.15** |
| | PR-BCD | 50.89±1.92 | _71.39±1.91_ | 65.69±1.35 | 70.49±1.20 | 41.90±0.94 | 70.40±1.28 | 57.10±1.37 | 66.80±1.89 | 67.59±1.28 | 69.69±2.00 | 67.20±1.08 | 65.10±1.51 | **74.10±1.22** |
| | CAMA | 52.19±1.33 | 68.40±0.66 | 59.19±0.75 | 64.50±1.20 | 57.40±0.48 | 69.20±0.98 | 54.30±1.35 | 67.60±1.50 | 55.99±1.41 | 67.60±2.11 | 61.10±1.45 | 56.70±1.62 | **75.90±0.99** |
| | Average | 55.29 | 66.33 | 63.36 | 67.96 | 50.06 | _70.53_ | 58.47 | 67.67 | 62.03 | 66.10 | 65.47 | 61.87 | **75.33** |
| **IMDB-M** | GradArgmax | 38.53±2.00 | 46.07±0.76 | 40.18±3.63 | 44.20±1.16 | 39.26±0.47 | 42.07±0.70 | 42.53±1.68 | 45.60±1.87 | 41.33±0.42 | _47.43±0.79_ | 38.20±0.67 | 44.40±0.94 | **48.47±1.12** |
| | PR-BCD | 35.00±1.31 | _46.00±1.46_ | 43.53±1.12 | 45.60±1.69 | 36.11±0.63 | 35.27±0.70 | 38.07±2.24 | 42.47±1.66 | 37.13±0.43 | 38.97±1.64 | 37.33±0.79 | 43.11±1.75 | **47.00±1.44** |
| | CAMA | 38.40±1.69 | _46.27±0.33_ | 42.80±0.88 | 46.00±0.94 | 37.99±1.69 | 44.47±0.99 | 41.00±1.50 | 45.67±2.12 | 41.13±1.23 | 43.56±0.60 | 39.73±1.74 | 38.87±1.46 | **48.13±2.44** |
| | Average | 37.31 | _46.11_ | 42.17 | 45.27 | 37.79 | 40.60 | 40.53 | 44.58 | 39.86 | 43.32 | 38.42 | 42.13 | **47.87** |
| **REDDIT-B** | GradArgmax | 40.24±0.51 | | 55.16±0.87 | 52.25±0.51 | 40.84±0.22 | _66.95±2.70_ | 66.40±0.49 | 64.40±1.88 | 62.90±0.76 | 59.80±0.78 | 54.00±0.32 | 57.35±0.74 | **67.35±0.55** |
| | PR-BCD | 51.82±1.09 | OOM | 51.96±0.57 | 57.06±1.55 | 55.05±1.55 | 54.85±1.94 | 51.65±0.32 | 52.05±1.78 | _64.20±1.94_ | 66.00±3.29 | 56.15±1.29 | 54.05±0.35 | **67.63±0.42** |
| | CAMA | 51.49±0.59 | | 58.84±0.95 | 62.65±0.90 | 54.95±0.57 | 66.50±3.02 | 48.10±0.92 | 67.85±1.90 | _69.90±0.49_ | 53.90±0.30 | 60.40±0.54 | 55.90±1.04 | **68.15±0.95** |
| | Average | 47.85 | OOM | 55.32 | 57.32 | 50.28 | 62.77 | 55.38 | 61.43 | _65.67_ | 59.90 | 56.85 | 55.77 | **67.71** |
| **COLLAB** | GradArgmax | 59.30±1.37 | 66.84±0.83 | 62.08±0.59 | _68.00±0.31_ | 51.49±0.50 | 62.86±1.19 | 52.88±0.45 | 54.83±1.12 | 58.68±0.39 | 62.62±0.74 | 62.98±0.52 | 61.10±1.00 | **68.08±0.78** |
| | PR-BCD | 46.74±0.70 | _67.00±1.13_ | 57.40±1.67 | 66.52±0.88 | 56.08±1.19 | 53.38±1.90 | 44.34±1.46 | 49.46±1.17 | 53.00±0.60 | 61.02±0.97 | 64.30±0.48 | 57.04±0.67 | **67.56±0.69** |
| | CAMA | 49.70±1.04 | **67.92±0.20** | 62.08±0.59 | 66.96±0.56 | 48.38±0.60 | 60.21±1.01 | 54.14±0.41 | 54.90±1.07 | 56.60±0.37 | 56.92±0.61 | 62.86±0.47 | 59.64±0.46 | 67.06±0.63 |
| | Average | 51.91 | _67.25_ | 60.52 | 67.16 | 51.98 | 58.82 | 50.45 | 53.06 | 56.09 | 60.19 | 63.38 | 59.26 | **67.57** |

1) *Structure Learning Based* methods, including GSR [67], GAR-NET [11], and GUARD [29]; 2) *Preprocessing Based* methods, including SVDGCN [13] and JaccardGCN [52]; 3) *Robust Aggregation Based* methods, including RGCN [70], Median-GCN [6], GN-NGuard [64], SoftMedian [17], and ElasticGCN [34]; and 4) *Adversarial Training Based* methods, represented by the GraphADV [58]. Details of these baselines can be found in Appendix D.2.

**Adversarial Attack Settings.** For graph classification, we evaluate the performance against three strong evasion attacks: PR-BCD [17], GradArgmax [10], and CAMA-subgraph [51]. For node classification, we evaluate six evasion attacks: 1) *Targeted Attacks*: PR-BCD [17], Nettack [71], and GR-BCD [17]; 2) *Non-targeted Attacks*: MinMax [32], DICE [72], and Random [32]. Further details on the attack methods and budget settings are provided in Appendix D.3.

**Hyperparameter Settings.** Details are provided in Appendix D.4.

## 5.2 Graph Classification Robustness

We evaluated the robustness of the graph classification task under three adversarial attacks across five datasets. Since the choice of classifier affects attack effectiveness, especially in graph classification due to pooling operations, it is crucial to standardize the model architecture. Simple changes like adding a linear layer or adjusting pooling strategies can reduce the impact of attacks. To ensure a fair comparison, we used a two-layer GCN with a linear layer and mean pooling for both the baselines and our proposed DiffSP. Each experiment was repeated 10 times, with results shown in Table 1.

**Result.** 1) DiffSP consistently outperforms all baselines under the PR-BCD attack and achieves the highest average robustness across all attacks on five datasets, with a notable 4.80% average improvement on the IMDB-BINARY dataset. 2) It's important to note that while baselines may excel against specific attacks, they often struggle with others. In contrast, DiffSP maintains consistent robustness across both datasets and attacks, thanks to its ability to learn clean distributions and purify adversarial graphs without relying on specific priors about the dataset or attack strategies.

## 5.3 Node Classification Robustness

We evaluate the robustness of DiffSP on the node classification task against three targeted and three non-targeted attacks across four datasets, using the same other settings as in the graph classification experiments. The results are presented in Table 2 and Table 3.

**Result.** We have two key observations: 1) DiffSP achieves the best average performance across both targeted and non-targeted attacks on all datasets, demonstrating its robust adaptability across diverse scenarios. 2) DiffSP performs particularly well under stronger attacks but is less effective against weaker ones like Random and DICE. This is because these attacks introduce numerous noisy edges, many of which do not exhibit distinctly adversarial characteristics. Instead, these edges are often plausible within the graph. Consequently, these additional perturbations can mislead DiffSP, making it harder to discern the correct information within the graph, leading the generated graph to deviate from the target clean graph.

## 5.4 Ablation Study

In this subsection, we analyze the effectiveness of DiffSP's two core components: 1) DiffSP (w/o LN), which excludes the LID-Driven Non-Isotropic Diffusion Mechanism, applying uniform noise to all nodes during the forward process; and 2) DiffSP (w/o TG), which excludes the Graph Transfer Entropy Guided Denoising Mechanism, thus removing guidance during the reverse process. We evaluate these variants on the IMDB-BINARY and COLLAB datasets under PR-BCD and GradArgmax attacks for graph classification and on the Cora and CiteSeer dataset under PR-BCD and MinMax attacks for node classification. The results are shown in Figure 3.

**Result.** DiffSP consistently outperforms the other variants. Without the LID-Driven Non-Isotropic Diffusion Mechanism, DiffSP (w/o LN) over-perturbs the valuable parts of the graph leading to degraded performance. Similarly, DiffSP (w/o TG) without the Transfer Entropy Guided Denoising Mechanism increases the uncertainty of generation, causing deviations from the target clean graph. These reduce the robustness against evasion attacks.

**Table 2: Accuracy score (% ± standard deviation) of *node classification* task on real-world datasets against *targeted attack*.**

| Dataset | Attack | GCN | GSR | GARNET | GUARD | SVD | Jaccard | RGCN | MedianGCN | GNNGuard | SoftMedian | ElasticGCN | GraphAT | DiffSP |
|---------|--------|-----|-----|--------|-------|-----|---------|------|-----------|----------|-----------|-----------|---------|--------|
| **Cora** | PR-BCD | 55.59±1.47 | 74.75±0.53 | 66.80±0.46 | 65.71±0.79 | 64.66±0.35 | 60.49±1.00 | 55.91±0.65 | 61.77±0.68 | 65.14±1.07 | 59.36±0.63 | 63.86±1.38 | 63.74±0.99 | 75.13±1.27 |
| | Nettack | 49.25±5.28 | 67.25±5.20 | 62.95±4.75 | 52.50±4.08 | 70.25±0.79 | 56.75±2.65 | 47.50±1.67 | 76.25±5.17 | 76.00±5.03 | 67.50±4.25 | 65.25±3.22 | 73.50±9.14 | 77.75±3.62 |
| | GR-BCD | 66.34±1.45 | 78.86±0.53 | 72.35±0.91 | 72.08±1.23 | 65.34±0.72 | 71.88±0.76 | 69.74±2.08 | 72.90±1.06 | 70.45±1.20 | 75.52±0.86 | 78.44±1.42 | 77.06±1.24 | 76.83±0.65 |
| | Average | 57.06 | 73.62 | 67.37 | 63.43 | 66.75 | 63.04 | 57.72 | 70.31 | 70.53 | 67.46 | 69.18 | 71.43 | 76.57 |
| **CiteSeer** | PR-BCD | 45.06±1.83 | 63.33±0.60 | 55.75±1.71 | 54.48±0.96 | 59.61±0.51 | 48.72±1.20 | 41.08±1.55 | 49.72±0.71 | 49.78±2.33 | 49.20±0.89 | 48.79±1.41 | 61.54±1.01 | 64.35±0.89 |
| | Nettack | 60.75±8.34 | 75.25±2.65 | 72.00±2.84 | 59.25±3.92 | 77.25±1.84 | 71.50±3.16 | 42.25±4.78 | 74.00±2.93 | 77.00±3.50 | 59.00±2.11 | 63.50±3.76 | 73.25±5.14 | 78.80±4.53 |
| | GR-BCD | 50.56±2.17 | 65.50±0.57 | 57.04±2.57 | 54.74±1.82 | 60.40±0.59 | 59.83±1.17 | 44.82±1.60 | 55.17±1.31 | 58.88±3.38 | 55.65±0.93 | 60.37±2.91 | 62.25±1.25 | 65.63±1.30 |
| | Average | 52.12 | 68.02 | 61.60 | 56.16 | 65.75 | 60.02 | 42.72 | 59.63 | 61.89 | 54.62 | 57.55 | 65.68 | 69.59 |
| **PolBlogs** | PR-BCD | 73.73±1.19 | 86.50±0.52 | 75.52±0.50 | 81.82±1.06 | 78.02±0.16 | 51.45±1.23 | 74.01±0.32 | 65.07±4.21 | 51.93±2.54 | 87.88±1.29 | 74.71±2.89 | 80.67±0.85 | 90.24±0.92 |
| | Nettack | 74.75±4.92 | 75.75±1.69 | 83.75±3.77 | 76.75±3.13 | 80.75±1.69 | 47.75±6.06 | 76.50±1.75 | 46.00±2.11 | 50.24±6.52 | 83.50±3.37 | 86.00±4.12 | 83.95±2.72 | 84.55±5.90 |
| | GR-BCD | 71.31±3.41 | 84.75±0.66 | 75.49±0.77 | 87.13±3.63 | 90.27±0.36 | 50.71±1.98 | 79.13±0.54 | 56.95±5.15 | 51.26±1.78 | 87.50±0.81 | 91.12±2.71 | 92.70±0.18 | 92.75±0.38 |
| | Average | 73.26 | 82.33 | 78.25 | 81.90 | 83.01 | 49.97 | 76.55 | 56.01 | 51.14 | 86.29 | 83.94 | 85.77 | 89.18 |
| **Photo** | PR-BCD | 65.35±2.48 | 73.81±1.90 | 77.58±1.93 | 84.14±3.75 | 80.04±1.13 | 66.13±2.82 | 63.79±11.99 | 79.75±0.96 | 65.62±2.63 | 76.84±1.46 | 76.21±1.89 | 78.72±2.13 | 84.78±1.82 |
| | Nettack | 83.70±5.16 | 83.75±4.12 | 88.00±3.07 | 84.25±2.65 | 82.75±5.45 | 84.00±5.43 | 75.50±3.07 | 86.50±3.16 | 87.50±5.77 | 88.75±1.32 | 83.00±3.29 | 87.25±12.30 | 87.75±4.32 |
| | GR-BCD | 69.11±7.85 | 84.84±2.29 | 85.27±1.57 | 82.15±2.24 | 83.74±1.11 | 76.24±2.98 | 68.60±7.28 | 84.23±1.49 | 79.20±1.80 | 79.69±1.19 | 83.94±0.95 | 87.49±1.26 | 87.58±0.58 |
| | Average | 72.72 | 80.80 | 83.62 | 83.51 | 82.18 | 75.46 | 69.30 | 83.49 | 77.44 | 81.76 | 81.05 | 84.48 | 86.70 |

**Table 3: Accuracy score (% ± standard deviation) of *node classification* task on real-world datasets against *non-targeted attack*.**

| Dataset | Attack | GCN | GSR | GARNET | GUARD | SVD | Jaccard | RGCN | MedianGCN | GNNGuard | SoftMedian | ElasticGCN | GraphAT | DiffSP |
|---------|--------|-----|-----|--------|-------|-----|---------|------|-----------|----------|-----------|-----------|---------|--------|
| **Cora** | MinMax | 59.91±2.60 | 67.80±2.18 | 65.68±0.58 | 61.62±2.85 | 64.75±0.96 | 64.43±2.48 | 62.49±2.19 | 56.35±3.34 | 63.63±2.40 | 74.53±0.70 | 17.05±5.33 | 63.35±2.60 | 75.00±1.12 |
| | DICE | 69.58±2.17 | 74.55±0.74 | 68.88±1.08 | 71.50±2.68 | 59.52±0.39 | 71.89±0.56 | 69.92±0.97 | 71.61±0.72 | 68.82±0.95 | 73.38±0.68 | 74.11±1.28 | 75.84±0.57 | 75.96±0.87 |
| | Random | 70.43±2.22 | 77.37±0.88 | 75.63±0.93 | 74.96±0.51 | 62.54±0.65 | 73.74±0.60 | 72.74±1.00 | 74.31±0.95 | 68.33±1.72 | 77.52±0.65 | 74.06±3.87 | 77.39±0.91 | 77.63±0.80 |
| | Average | 66.64 | 73.24 | 70.06 | 69.36 | 62.27 | 70.02 | 68.38 | 67.42 | 66.93 | 75.14 | 55.07 | 72.19 | 76.20 |
| **CiteSeer** | MinMax | 52.07±6.63 | 54.74±4.92 | 59.00±2.35 | 58.02±1.44 | 35.83±1.89 | 56.65±3.81 | 42.85±7.72 | 53.39±3.44 | 57.98±2.97 | 60.84±1.40 | 17.05±5.33 | 61.54±3.70 | 61.59±1.10 |
| | DICE | 57.46±1.63 | 62.48±1.08 | 55.59±3.01 | 62.19±0.99 | 57.33±0.49 | 63.00±0.87 | 50.88±1.59 | 59.95±0.97 | 58.85±3.22 | 59.85±0.81 | 60.30±1.46 | 65.28±0.81 | 65.43±0.70 |
| | Random | 56.19±3.08 | 64.01±1.08 | 56.34±3.70 | 62.47±0.68 | 54.54±0.62 | 64.20±0.46 | 50.13±1.95 | 60.60±0.81 | 61.51±3.32 | 58.66±1.49 | 58.00±3.61 | 64.94±1.12 | 66.78±0.54 |
| | Average | 55.24 | 60.41 | 56.98 | 60.89 | 49.23 | 61.28 | 47.95 | 57.98 | 59.45 | 59.78 | 45.12 | 63.92 | 64.60 |
| **PolBlogs** | MinMax | 86.96±0.43 | 88.56±0.82 | 87.85±0.19 | 89.51±0.85 | 87.11±0.32 | 51.01±1.75 | 87.04±0.19 | 87.95±4.81 | 50.32±1.19 | 88.76±0.37 | 87.33±0.62 | 88.32±0.35 | 89.52±3.08 |
| | DICE | 76.52±2.76 | 80.75±4.72 | 85.05±1.01 | 83.76±0.78 | 82.84±0.20 | 50.27±1.91 | 81.50±0.44 | 74.19±3.02 | 50.79±1.59 | 86.47±0.45 | 82.40±2.24 | 87.39±0.44 | 88.85±1.32 |
| | Random | 83.24±5.81 | 87.81±1.03 | 83.42±1.59 | 87.48±1.51 | 85.59±0.31 | 51.02±1.75 | 85.46±0.40 | 83.57±2.71 | 50.28±1.13 | 90.35±0.56 | 49.50±2.20 | 90.50±0.56 | 92.61±0.93 |
| | Average | 82.24 | 85.71 | 85.44 | 86.92 | 85.18 | 50.77 | 84.67 | 81.90 | 50.46 | 88.53 | 73.08 | 88.74 | 90.33 |
| **Photo** | MinMax | 73.12±3.17 | 76.36±3.09 | 81.75±1.91 | 75.89±3.28 | 69.92±5.50 | 74.20±3.94 | 87.04±0.19 | 67.43±4.31 | 71.44±6.66 | 85.23±2.12 | 8.56±3.24 | 81.70±2.48 | 88.51±0.61 |
| | DICE | 84.60±1.17 | 82.52±1.66 | 85.43±0.92 | 82.92±1.27 | 76.42±1.39 | 83.20±1.44 | 81.57±0.44 | 82.83±2.45 | 83.87±1.19 | 84.72±0.90 | 81.86±3.61 | 87.22±1.13 | 83.52±1.19 |
| | Random | 85.38±1.76 | 83.62±2.91 | 84.12±3.95 | 85.49±1.55 | 79.13±2.84 | 83.37±1.93 | 86.87±2.89 | 84.07±2.52 | 83.24±4.83 | 85.95±1.06 | 75.32±2.38 | 86.23±2.26 | 84.60±0.46 |
| | Average | 81.03 | 80.83 | 83.77 | 81.43 | 75.16 | 80.26 | 85.16 | 78.11 | 79.52 | 85.30 | 55.25 | 85.05 | 85.54 |

## 5.5 Study on Cross-Dataset Generalization

In this section, we assess DiffSP's ability to generalize across datasets. The goal is to determine whether DiffSP effectively learns the predictive patterns of clean graphs. We train DiffSP on IMDB-BINARY and use the trained model to purify graphs on IMDB-MULTI under adversarial attacks, and vice versa. Results are shown in Table 4.

**Result.** As shown in Table 4, DiffSP trained on different datasets, still demonstrates strong robustness compared to GCN trained and tested on the same dataset. Furthermore, DiffSP exhibits only a small performance gap compared to when it is trained and tested on the same dataset directly. These results highlight DiffSP's ability to learn the underlying clean distribution of a category of data and capture predictive patterns that generalize across diverse datasets.

## 5.6 Study on Purification Steps

We evaluate both classification accuracy and purification time as the number of diffusion steps varies. For graph classification on the IMDB-BINARY dataset, we adjust the diffusion steps from 1 to 9 and assess the performance under the GradArgMax, PR-BCD, and CAMA-Subgraph attacks. Similarly, for node classification on the Cora dataset, we vary the diffusion steps from 1 to 12, evaluating

**Table 4: Accuracy (% ± standard deviation) across datasets. (B→B) indicates the model is both trained and tested on IMDB-BINARY, while (M→B) indicates the model is trained on IMDB-MULTI but tested on IMDB-BINARY.**

| Attack | GCN (B→B) | DiffSP (B→B) | DiffSP (M→B) | GCN (B→B) | DiffSP (M→M) | DiffSP (B→M) |
|--------|-----------|--------------|--------------|-----------|--------------|--------------|
| **PR-BCD** | 50.90±1.92 | 74.10±1.29 | 73.90±2.02 | 35.00±1.31 | 47.00±1.44 | 45.33±0.99 |
| **GradArgmax** | 62.80±1.08 | 76.00±1.15 | 75.00±1.70 | 38.53±2.00 | 48.47±1.12 | 47.60±1.41 |
| **CAMA** | 52.20±1.33 | 75.90±0.99 | 75.10±1.37 | 38.40±1.69 | 48.13±2.44 | 47.47±0.88 |

the results under the GR-BCD, PR-BCD, and MinMax attacks. The results are shown in Figure 4.

**Result.** We observe that all-time step settings demonstrate the ability to effectively purify adversarial graphs. At smaller time steps, the overall trend shows increasing accuracy as the number of diffusion steps increases. This is likely because fewer time steps do not introduce enough noise to sufficiently suppress the adversarial information in the graph. As the diffusion steps increase, we do not see a significant decline in performance. This stability can be attributed to our LID-Driven Non-Isotropic Diffusion Mechanism, which minimizes over-perturbation of the clean graph parts. Additionally, we found that the time required for purifying increased linearly, which aligns with our expectations.

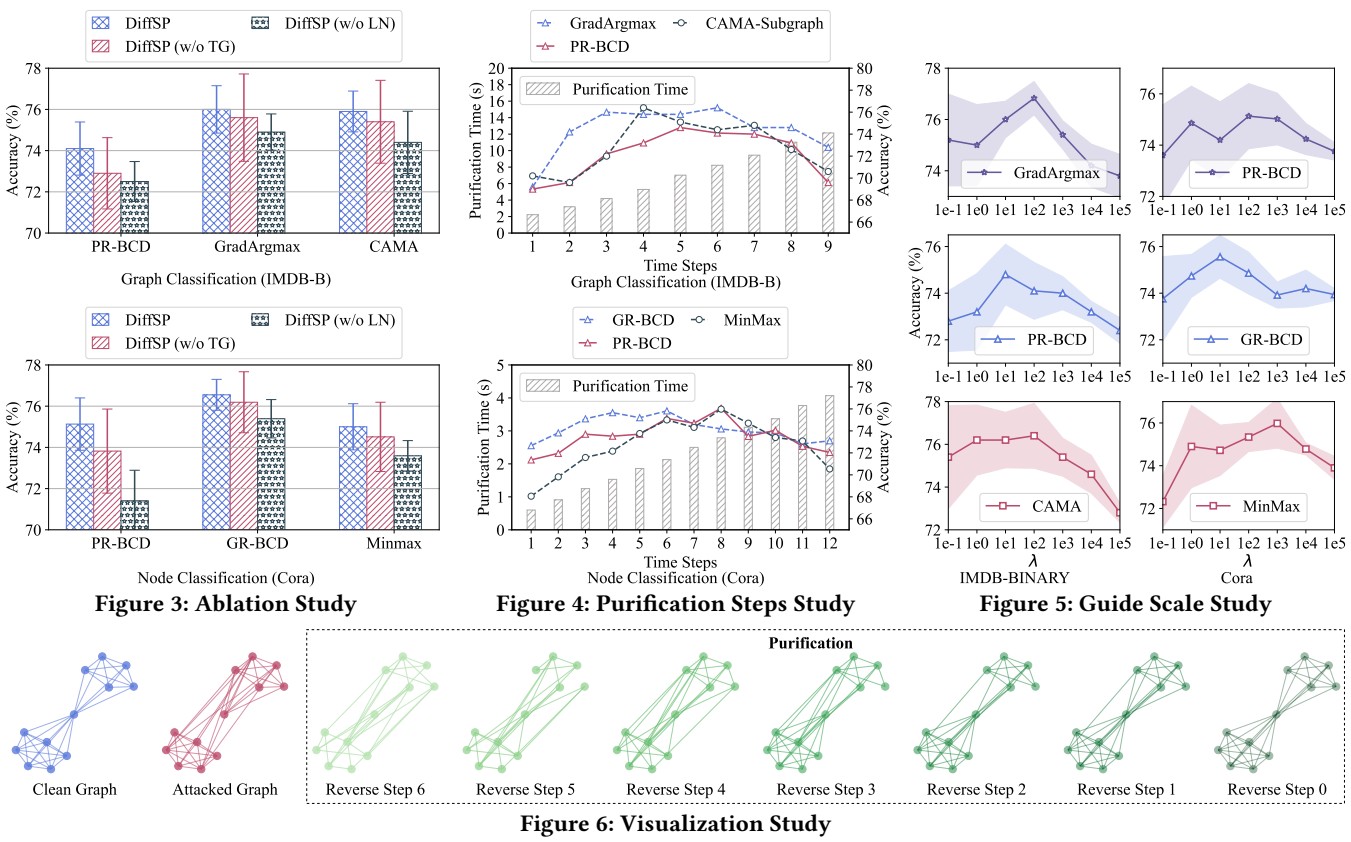

Figure 3: Ablation Study

Figure 4: Purification Steps Study

Figure 5: Guide Scale Study

Figure 6: Visualization Study

## 5.7 Study on Scale of Graph Transfer Entropy

To analyze the impact of the graph transfer entropy guidance scale $\lambda$, we vary $\lambda$ from $1e^{-1}$ to $1e^5$ and evaluate the performance. The results are presented in Figure 5. For graph classification, experiments are conducted on the IMDB-BINARY dataset under the GradArgmax, PR-BCD, and CAMA-Subgraph attacks. For node classification, experiments are performed on the Cora dataset under the PR-BCD, GR-BCD, and MinMax attacks.

**Result.** The results show that smaller values of $\lambda$ have minimal effect on accuracy. However, they reduce the stability of the purification during the reverse denoising process, leading to a higher standard deviation. This instability arises because the Graph Transfer Entropy Guided Denoising Mechanism is less effective at reducing uncertainty and guiding the generation process when $\lambda$ is too small. On the other hand, very large $\lambda$ values decrease accuracy by overemphasizing guidance, causing the model to reintroduce adversarial information into the generated graph structure.

## 5.8 Graph Purification Visualization

We visualize snapshots of different purification time steps on the IMDB-BINARY dataset using NetworkX [19], as shown in Figure 6.

**Result.** The visualization process demonstrates that DiffSP has mastered the ability to generate clean graphs. When faced with an attacked graph, DiffSP first injects noise to obscure the adversarial information, then applies its captured predictive patterns to remove both the adversarial information and the noise during the reverse denoising process, thereby achieving graph purification.

More experiments and analyses are provided in Appendix E.

## 6 Conclusion

Under adversarial evasion attacks, most existing methods rely on heuristic priors about the dataset or attack strategies to enhance robustness, which limits their effectiveness in real-world scenarios where these priors not universally hold. To address this, we propose a novel framework named DiffSP, which achieves prior-free structure purification to ensure robust graph learning across diverse evasion attacks and datasets. DiffSP innovatively adopts the graph diffusion model to learn the clean graph distribution during training and purify the attacked graph under the direction of captured predictive patterns during the test phase. To precisely denoise the attacked graph without disrupting the clean structure, we design an LID-Driven Non-Isotropic Diffusion Mechanism to inject varying levels of noise into each node based on their adversarial degree. To align the semantic information between the generated graph and the target clean graph, we design a Graph Transfer Entropy Guided Denoising Mechanism to reduce generation uncertainty and guide the generation direction. Extensive experimental results demonstrate that DiffSP enhances the robustness of graph learning in various scenarios. In future work, we aim to incorporate feature-based attack experiments and optimize the time complexity of DiffSP. Additionally, we plan to improve our proposed graph entropy tool and explore its application to address a wider range of challenges in graph learning and other related tasks. Details about the limitations and future directions can be found in Appendix F.

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

# A  Proof and Derivation

## A.1  Proof of Proposition 1

We first restate Propostition 1.

PROPOSITION 1. *For each edge at time $t$, the adjacency matrix is updated as $\mathbf{A}_{ij}^{(t)} = \mathbf{A}_{ij}(\bar{\mathbf{Q}}_{\mathbf{A}}'^{(t)})_{ij}$, where the non-isotropic transition matrix is $(\bar{\mathbf{Q}}_{\mathbf{A}}'^{(t)})_{ij} = \bar{\alpha}^{(t)}\mathbf{I} + (\mathbf{\Lambda}_{\mathbf{A}})_{ij}(1-\bar{\alpha})\mathbf{1}\mathbf{m}_{\mathbf{A}}^T$. There exists a unique time $\hat{t}(\mathbf{A}_{ij}) \in [0, T]$ such that $(\bar{\mathbf{Q}}_{\mathbf{A}}'^{(t)})_{ij} \Leftrightarrow (\bar{\mathbf{Q}}_{\mathbf{A}}^{\hat{t}(\mathbf{A}_{ij})})_{ij}$, where:*

$$\hat{t}(\mathbf{A}_{ij}) = T\left(\frac{2(1+s)}{\pi}\cos^{-1}\left(\sqrt{\frac{\bar{\alpha}^{(t)}}{[\mathbf{\Lambda}(\mathbf{A})_{ij}(1-\bar{\alpha}^{(t)})+\bar{\alpha}^{(t)}]}}\right) - s\right).$$

PROOF. $\bar{\mathbf{Q}}_{\mathbf{A}}^{(t)} = \bar{\alpha}^{(t)}\mathbf{I} + (1-\bar{\alpha}^{(t)})\mathbf{1}\mathbf{m}_{\mathbf{A}}^T$ indicates the degree of noise added to the adjacency matrix $\mathbf{A}$ at time step $t$. Let $\text{SNR}_{\bar{\mathbf{Q}}_{\mathbf{A}}}(t)$ denotes the signal-to-noise of $\bar{\mathbf{Q}}_{\mathbf{A}}^{(t)}$ at time step $t$, we have:

$$\text{SNR}_{\bar{\mathbf{Q}}_{\mathbf{A}}}(t) = \frac{1-\bar{\alpha}^{(t)}}{\bar{\alpha}^{(t)}}. \tag{A.1}$$

Such that:

$$(\bar{\mathbf{Q}}_{\mathbf{A}}'^{(t)})_{ij} \Leftrightarrow (\bar{\mathbf{Q}}_{\mathbf{A}}^{\hat{t}(\mathbf{A}_{ij})})_{ij} \tag{A.2}$$

$$\Rightarrow \text{SNR}_{\bar{\mathbf{Q}}_{\mathbf{A}}'}(t) = \text{SNR}_{\bar{\mathbf{Q}}_{\mathbf{A}}}(\hat{t}(\mathbf{A}_{ij})) \tag{A.3}$$

$$\Rightarrow \frac{(\mathbf{\Lambda}_{\mathbf{A}})_{ij}(1-\bar{\alpha}^{(t)})}{\bar{\alpha}^{(t)}} = \frac{1-\bar{\alpha}^{(t')}}{\bar{\alpha}^{(t')}}. \tag{A.4}$$

We first prove that for each time step $t$, there exists and only exists one $t'$ that satisfies $(\bar{\mathbf{Q}}_{\mathbf{A}}'^{(t)})_{ij} \Leftrightarrow (\bar{\mathbf{Q}}_{\mathbf{A}}^{\hat{t}(\mathbf{A}_{ij})})_{ij}$. Left $g(t') = \frac{(\mathbf{\Lambda}_{\mathbf{A}})_{ij}(1-\bar{\alpha}^{(t)})}{\bar{\alpha}^{(t)}} - \frac{1-\bar{\alpha}^{(t')}}{\bar{\alpha}^{(t')}}$ represents the function of $t' \in [0, T]$. $\bar{\alpha}^{(t)} = \cos^2\left(\frac{t/T+s}{1+s}\cdot\frac{\pi}{2}\right)$ is the scheduler with a small constant $s$. We have $\alpha^{(0)} = \cos^2\left(\frac{0+s}{1+s}\cdot\frac{\pi}{2}\right) \approx \cos^2(0) = 0$, and $\alpha^{(T)} = \cos^2\left(\frac{1+s}{1+s}\cdot\frac{\pi}{2}\right) = \cos^2(\frac{\pi}{2}) = 1$. It is known that $(1-\bar{\alpha})$ monotonically decreasing over the domain, while $\bar{\alpha}$ monotonically increasing, with $1-\bar{\alpha} > 0$ and $\bar{\alpha} > 0$. Therefore, $g(t')$ is a monotonic function over the domain. So we achieve:

$$g(0) = \frac{(\mathbf{\Lambda}_{\mathbf{A}})_{ij}(1-\bar{\alpha}^{(t)})}{\bar{\alpha}^{(t)}} - 0 > 0. \tag{A.5}$$

Having $\mathbf{\Lambda}(\mathbf{A})_{ij} \in [0, 1]$ indicates the node adversarial score, we can then derive the following:

$$g(T) = \frac{(\mathbf{\Lambda}_{\mathbf{A}})_{ij}(1-\bar{\alpha}^{(t)})}{\bar{\alpha}^{(t)}} - 1 \tag{A.6}$$

$$< \frac{(1-\bar{\alpha}^{(t)})}{\bar{\alpha}^{(t)}} - 1 \tag{A.7}$$

$$< 0. \tag{A.8}$$

Thus, we have $g(0)g(T) < 0$, and since $g(t')$ is a monotonically decreasing function, the intermediate value theorem guarantees that there exists exactly one $t'_0 \in [0, T]$ satisfies $g(t'_0) = 0$. By setting

$g(t') = 0$, we obtain:

$$\mathbf{\Lambda}(\mathbf{A})_{ij}\bar{\alpha}^{(t')}(1-\bar{\alpha}^{(t)}) = \bar{\alpha}^{(t)}(1-\bar{\alpha}^{(t')}) \tag{A.9}$$

$$\Rightarrow \bar{\alpha}^{(t')}[\mathbf{\Lambda}(\mathbf{A})_{ij}(1-\bar{\alpha}^{(t)})+\bar{\alpha}^{(t)}] = \bar{\alpha}^{(t)} \tag{A.10}$$

$$\Rightarrow \bar{\alpha}^{(t')} = \frac{\bar{\alpha}^{(t)}}{[\mathbf{\Lambda}(\mathbf{A})_{ij}(1-\bar{\alpha}^{(t)})+\bar{\alpha}^{(t)}]} \tag{A.11}$$

$$\Rightarrow t' = T\left(\frac{2(1+s)}{\pi}\cos^{-1}\left(\sqrt{\frac{\bar{\alpha}^{(t)}}{[\mathbf{\Lambda}(\mathbf{A})_{ij}(1-\bar{\alpha}^{(t)})+\bar{\alpha}^{(t)}]}}\right) - s\right). \tag{A.12}$$

This concludes the proof of the proposition. □

## A.2  Graph Transfer Entropy Derivation

We first restate Eq. (7).

$$I(\hat{G}^{t-1}; G_{\text{adv}}|\hat{G}_t) = H(\hat{G}^{(t-1)}|\hat{G}^{(t)}) - H(\hat{G}^{(t-1)}|\hat{G}^{(t)}, G_{\text{adv}}).$$

According to the definition of mutual information:

$$I(\hat{G}^{t-1}; G_{\text{adv}}|\hat{G}_t) \tag{A.13}$$

$$= H(\hat{G}^{(t-1)}|\hat{G}^{(t)}) - H(\hat{G}^{(t-1)}|\hat{G}^{(t)}, G_{\text{adv}}) \tag{A.14}$$

$$= \frac{H(\hat{G}^{(t-1)}, \hat{G}^{(t)})}{H(\hat{G}^{(t)})} - \frac{H(\hat{G}^{(t-1)}, \hat{G}^{(t)}, G_{\text{adv}})}{H(\hat{G}^{(t)}, G_{\text{adv}})}. \tag{A.15}$$

Then combined with Eq. (9), we have:

$$I(\hat{G}^{t-1}; G_{\text{adv}}|\hat{G}_t) \tag{A.16}$$

$$= S_\alpha\left(\frac{\hat{\mathbf{K}}^{(t-1)} \odot \hat{\mathbf{K}}^{(t)}}{\text{tr}(\hat{\mathbf{K}}^{(t-1)} \odot \hat{\mathbf{K}}^{(t)})}\right)/S_\alpha(\hat{\mathbf{K}}^{(t)}) \tag{A.17}$$

$$- S_\alpha\left(\frac{\hat{\mathbf{K}}^{(t-1)} \odot \hat{\mathbf{K}}^{(t)} \odot \mathbf{K}_{\text{adv}}}{\text{tr}(\hat{\mathbf{K}}^{(t-1)} \odot \hat{\mathbf{K}}^{(t)} \odot \mathbf{K}_{\text{adv}})}\right)/S_\alpha\left(\frac{\hat{\mathbf{K}}^{(t)} \odot \mathbf{K}_{\text{adv}}}{\text{tr}(\hat{\mathbf{K}}^{(t)} \odot \mathbf{K}_{\text{adv}})}\right), \tag{A.18}$$

where $S_\alpha(\cdot)$ is the graph entropy calculated according to Eq. (8) and $\hat{\mathbf{K}}^{(t-1)}, \hat{\mathbf{K}}^{(t)}, \mathbf{K}_{\text{adv}}$ is the Gram matrix of $\hat{\mathbf{A}}^{(t-1)}, \hat{\mathbf{A}}^{(t)}, \mathbf{A}_{\text{adv}}$.

# B  Detailed Understanding of the Proposed Graph Transfer Entropy

In this subsection, we further elaborate on the understanding of our graph entropy estimation method in Eq. (8). After message passing, the set of node representations $\mathbf{Z}$ can be treated as variables that capture both structural and node feature neighborhood information. The normalized Gram matrix $\hat{\mathbf{K}}$, obtained by applying a positive definite kernel on all pairs of $z$, measures the neighborhood similarity between each pair of nodes, taking into account both node features and neighboring structures. Let $\lambda_i(\hat{\mathbf{K}})$ be the eigenvalue of $\hat{\mathbf{K}}$ with eigenvector $\mathbf{x}_i$. Then we have:

$$\hat{\mathbf{K}}^2 = \hat{\mathbf{K}}(\hat{\mathbf{K}}\mathbf{x}_i) = \hat{\mathbf{K}}(\lambda_i(\hat{\mathbf{K}})\mathbf{x}_i) = \lambda_i(\hat{\mathbf{K}})\hat{\mathbf{K}}\mathbf{x}_i = \lambda_i^2(\hat{\mathbf{K}})\mathbf{x}_i. \tag{B.19}$$

Thus we achieve:

$$\sum_{i=1}^{n}\lambda_i^\alpha(\hat{\mathbf{K}}) = \sum_{i=1}^{n}\lambda_i(\hat{\mathbf{K}}^\alpha). \tag{B.20}$$

Since the sum of all eigenvalues of a matrix is the trace of the matrix, the graph entropy is determined by the trace of $\hat{\mathbf{K}}^\alpha$. By setting $\alpha = 2$, $\hat{\mathbf{K}}_{ii}^2$ describes the similarity of node $i$ with all other nodes, considering both node features and neighboring structures. When

$\alpha = 2$, the graph entropy can be expressed as: $H(G) = -\log \text{tr}(\hat{\mathbf{K}}^2)$ Therefore a lower graph entropy indicates a graph with a stronger community structure, while a higher graph entropy suggests a more chaotic graph structure with less regularity. So maximizing the transfer entropy $I(\hat{G}^{(t-1)}; G_{\text{adv}} | \hat{G}^{(t)})$ actually encourage the community structure of $\hat{G}^{(t-1)}$ move towards $G_{\text{adv}}$.

## C  Computational Complexity Analysis

The overall time complexity is $O(N^2)$, where $N$ represents the number of nodes. Specifically, the graph diffusion purification model has a complexity of $O(TN^2)$. The LID-Driven Non-Isotropic Diffusion Module has a complexity of $O(N)$, and the Transfer Entropy Guided Diffusion Module has a complexity of $O(N^2)$. Therefore, the overall time complexity of the purification process is $O(TN^2) + O(N) + O(N^2) = O(TN^2)$. Since $T \ll N^2$ in our case, the overall time complexity is $O(N^2)$. This is consistent with most graph diffusion models [31, 39, 49] and robust GNNs [13, 26, 67].

## D  Experiment Details

### D.1  Dataset Details

*D.1.1  Graph Classification Datasets.* We use the following five real-world datasets to evaluate the robustness of DiffSP on the graph classification task. All the dataset is obtained from PyG TUDataset[2]

- **MUTAG** [24] contains graphs of small molecules, with nodes as atoms and edges representing chemical bonds. Labels indicate molecular toxicity.
- **IMDB-BINARY** [24] consists of movie-related graphs, where nodes are individuals, and edges represent relationships. Labels classify the movie as Action or Romance.
- **IMDB-MULTI** [24] is similar, but edges connect nodes across three genres: Comedy, Romance, and Sci-Fi, with corresponding labels.
- **REDDIT-BINARY** [24] features user discussion graphs from Reddit, with edges indicating responses. Graphs are labeled as either question-answer or discussion-based.
- **COLLAB** [24] consists of collaboration networks, where nodes are researchers, and edges represent collaborations. Labels identify the research field: High Energy Physics, Condensed Matter Physics, or Astro Physics.

Statistics of the graph classification datasets are in Table D.1.

*D.1.2  Node Classification Datasets.* We use the following four real-world datasets to evaluate the robustness of DiffSP on the node classification task.

- **Cora** [59] is a citation network where nodes represent publications, with binary word vectors as features. Edges indicate citation relationships.
- **CiteSeer** [59] is another citation network, similar to Cora, with nodes representing research papers and edges denoting citation links.
- **PolBlogs** [1] is a political blog network, where edges are hyperlinks between blogs. Nodes are labeled by political affiliation: liberal or conservative.

---

[2] https://pytorch-geometric.readthedocs.io/en/latest/generated/torch_geometric.datasets.TUDataset.html

### Table D.1: Statistics for graph classification datasets

| Dataset | #graph | #avg. node | #avg. edge | #feature | #class |
|---|---|---|---|---|---|
| MUTAG | 188 | 17.9 | 39.6 | 7 | 2 |
| IMDB-BINARY | 100 | 19.8 | 193.1 | / | 2 |
| IMDB-MULTI | 1500 | 13.0 | 65.9 | / | 3 |
| REDDIT-BINARY | 2000 | 429.6 | 995.5 | / | 2 |
| COLLAB | 5000 | 74.5 | 4914.4 | / | 2 |

### Table D.2: Statistics for node classification datasets

| Dataset | #node | #edge | #feature | #class |
|---|---|---|---|---|
| Cora | 2708 | 10556 | 1433 | 7 |
| CiteSeer | 3327 | 9104 | 3703 | 2 |
| PolBlogs | 1490 | 19025 | / | 2 |
| Photo | 7487 | 119043 | 745 | 8 |

- **Photo** [41] is a co-purchase network from Amazon, where nodes are products, edges represent frequent co-purchases, and features are bag-of-words from product reviews. Class labels indicate product categories.

The statistics of the graph classification datasets are given in Table D.2. Cora and CiteSeer is obtained from PyG Planetoid[3]. PolBlogs is obtained from PyG PolBlogs[4]. Photo is obtained from PyG Amazon[5].

### D.2  Description of Baselines

*D.2.1  Graph Classification Baselines.*

- **IDGL** [9] iteratively refines graph structures and embeddings for robust learning in noisy graphs.
- **GraphCL** [60] maximizes agreement between augmented graph views via contrastive loss.
- **VIB-GSL** [47] applies the Information Bottleneck to learn task-relevant graph structures.
- **G-Mixup** [20] generates synthetic graphs by mixing graphons to enhance generalization.
- **SEP** [53] minimizes structural entropy for optimized graph pooling.
- **MGRL** [36] addresses semantic bias and confidence collapse with instance-view consistency and class-view learning.
- **SCGCN** [68] ensures robustness with temporal and perturbation stability.
- **HGP-SL** [65] combines pooling and structure learning to preserve key substructures.
- **SubGattPool** [4] uses subgraph attention and hierarchical pooling for robust classification.
- **DIR** [55] identifies stable causal structures via interventional separation.
- **VGIB** [61] filters irrelevant nodes through noise injection for improved subgraph recognition.

In our implementation, since the authors of MGRL and Sub-GattPool have not provided open access to their code, we reproduced their methods based on the descriptions in their papers. The

---

[3] https://pytorch-geometric.readthedocs.io/en/latest/generated/torch_geometric.datasets.Planetoid.html#torch_geometric.datasets.Planetoid

[4] https://pytorch-geometric.readthedocs.io/en/latest/generated/torch_geometric.datasets.PolBlogs.html#torch_geometric.datasets.PolBlogs

[5] https://pytorch-geometric.readthedocs.io/en/latest/generated/torch_geometric.datasets.Amazon.html#torch_geometric.datasets.Amazon

implementations of other baselines can be found at the following URLs:

- **IDGL**: https://github.com/hugochan/IDGL
- **GraphCL**: https://github.com/Shen-Lab/GraphCL
- **VIB-GSL**: https://github.com/VIB-GSL/VIB-GSL
- **G-Mixup**: https://github.com/ahxt/g-mixup
- **SEP**: https://github.com/Wu-Junran/SEP
- **SCGCN**: https://github.com/DataLab-atom/temp
- **HGP-SL**: https://github.com/cszhangzhen/HGP-SL
- **DIR**: https://github.com/Wuyxin/DIR-GNN
- **VGIB**: https://github.com/Samyu0304/VGIB

*D.2.2    Node Classification Baselines.*

- **GSR** [67] refines graph structures via a pretrain-finetune pipeline using multi-view contrastive learning to estimate and adjust edge probabilities.
- **GARNET** [11] improves GNN robustness by using spectral embedding and probabilistic models to filter adversarial edges.
- **GUARD** [29] creates a universal defensive patch to remove adversarial edges, providing node-agnostic, scalable protection.
- **SVDGCN** [13] applies Truncated SVD preprocessing with a two-layer GCN.
- **JaccardGCN** [52] drops dissimilar edges in the graph before training a GCN.
- **RGCN** [70] models node features as Gaussian distributions, using variance-based attention for robustness.
- **Median-GCN** [6] improves robustness by using median aggregation instead of the weighted mean.
- **GNNGuard** [64] defends GNNs by pruning suspicious edges through neighbor importance estimation.
- **SoftMedian** [17] filters outliers by applying a weighted mean based on distance from the median to defend against adversarial noise.
- **ElasticGNN** [34] combines 1-based and 2-based smoothing, balancing global and local smoothness for better defense.
- **GraphADV** [58] boosts robustness through adversarial training with gradient-based topology attacks.

The implementations of these node classification baselines can be found at the following URLs:

- **GSR**: https://github.com/andyjzhao/WSDM23-GSR
- **GARNET**: https://github.com/cornell-zhang/GARNET
- **GUARD**: https://github.com/EdisonLeeeee/GUARD
- **SVD**: https://github.com/DSE-MSU/DeepRobust/blob/master/deeprobust/graph/defense/gcn_preprocess.py
- **Jaccard**: https://github.com/DSE-MSU/DeepRobust/blob/master/deeprobust/graph/defense/gcn_preprocess.py
- **RGCN**: https://github.com/DSE-MSU/DeepRobust/blob/master/deeprobust/graph/defense/r_gcn.py
- **Median-GCN**: https://github.com/DSE-MSU/DeepRobust/blob/master/deeprobust/graph/defense/median_gcn.py
- **GNNGuard**: https://github.com/mims-harvard/GNNGuard
- **SoftMedian**: https://github.com/sigeisler/robustness_of_gnns_at_scale
- **ElasticGCN**: https://github.com/lxiaorui/ElasticGNN
- **GraphADT**: https://github.com/KaidiXu/GCN_ADV_Train

## D.3    Attack Setting Details

*D.3.1    Graph Classification Attack Settings.* For graph classification attacks, we use the following three attack methods:

- **GradArgmax** [10] greedily selects edges for perturbation based on the gradient of each node pair.
- **PR-BCD** [17] performs sparsity-aware first-order optimization attacks using randomized block coordinate descent, enabling efficient attacks on large-scale graphs.
- **CAMA-Subgraph** [51] enhances adversarial attacks in graph classification by targeting critical subgraphs. It identifies top-ranked nodes via a Class Activation Mapping (CAM) framework and perturbs edges within these subgraphs to craft more precise adversarial examples.

Note that, as the authors of CAMA-Subgraph have not provided open access to their code, we reproduced their method based on the descriptions in their papers. The reproduced code is available in our repository. For the implementation of other baselines, we used code from the following URLs:

- **GradArgmax**: https://github.com/xingchenwan/grabnel/blob/main/src/attack/grad_arg_max.py
- **PR-BCD**: https://github.com/pyg-team/pytorch_geometric/blob/master/torch_geometric/contrib/nn/models/rbcd_attack.py

For all graphs in the dataset, we set 20% of the total number of edges as the attack budget. We use a two-layer GCN followed by a mean pooling layer and a linear layer as the surrogate model, which shares the same architecture as the classifier for all baselines.

*D.3.2    Node Classification Attack Settings.* For targeted node classification attacks, we use the following three attack methods:

- **PR-BCD** [17] performs the same attack as in graph classification but targets a different task.
- **Nettack** [71] incrementally modifies key edges or features to maximize the difference in log probabilities between correct and incorrect classes, while preserving the graph's core properties, such as the degree distribution.
- **GR-BCD** [17] is similar to PR-BCD but flips edges greedily based on the gradient concerning the adjacency matrix.

The implements of these attacks can be found from the following URLs:

- **PR-BCD**: https://github.com/pyg-team/pytorch_geometric/blob/master/torch_geometric/contrib/nn/models/rbcd_attack.py
- **Nettack**: https://github.com/DSE-MSU/DeepRobust/blob/master/deeprobust/graph/targeted_attack/nettack.py
- **GR-BCD**: https://github.com/pyg-team/pytorch_geometric/blob/master/torch_geometric/contrib/nn/models/rbcd_attack.py

For all datasets, we set 10% of the total number of edges as the attack budget for both PR-BCD and GR-BCD. For Nettack, following the settings from deeprobust [32], we select 40 nodes from the test set to attack with a budget of 5 edges and evaluate accuracy. These 40 nodes include 1) 10 nodes with the highest classification margin (clearly correctly classified), 2) 10 nodes with the lowest margin (still correctly classified), and 3) 20 randomly selected nodes.

For non-targeted node classification attacks, we use the following three attack methods:

### Table D.3: Hyperparameter settings

| Hyperparameter | MT | IB | IM | RB | CL | Cora | CiteSeer | PolBlogs | Photo |
|---|---|---|---|---|---|---|---|---|---|
| $k$ | 4 | 6 | 6 | 8 | 8 | 7 | 8 | 8 | 8 |
| $\lambda$ | 1e1 | 1e2 | 1e3 | 1e3 | 1e3 | 1e3 | 1e3 | 1e3 | 1e3 |
| purification steps | 4 | 6 | 5 | 6 | 4 | 6 | 6 | 6 | 6 |

- **MinMax** [32] generates adversarial perturbations by solving a min-max optimization. The outer step finds optimal edge perturbations, while the inner step retrains the GNN to adapt.
- **DICE** [72] removes edges between same-class nodes and inserts edges between nodes of different classes.
- **Random** [32] randomly adds edges to the input graph.

The implements of theses attacks can be found in the following URLs:

- **MinMax**: https://github.com/DSE-MSU/DeepRobust/blob/master/deeprobust/graph/global_attack/topology_attack.py
- **DICE**: https://github.com/DSE-MSU/DeepRobust/blob/master/deeprobust/graph/global_attack/dice.py
- **Random**: https://github.com/DSE-MSU/DeepRobust/blob/master/deeprobust/graph/global_attack/random_attack.py

For MinMax, DICE, and Random attacks, we set the attack budget to 10%, 20%, and 30% of the total number of edges, respectively, for all datasets.

### D.4 Implement Details

For graph classification, we randomly split the dataset into 8:1:1 for training, validation, and testing. For datasets without node features, we use normalized node degrees as features, following the approach in [47]. The testing set is subjected to adversarial attacks. Our classifier consists of a two-layer Graph Convolutional Network (GCN) followed by a mean pooling layer and a linear layer. Both the diffusion model of DiffSP and the classifier are trained on the training graphs, with their performance evaluated on the attacked testing set. For node classification, we use the transductive setting with a 1:1:8 random split for training, validation, and testing. The classifier comprises a two-layer GCN followed by a linear layer. During training, we sample batches of subgraphs, consistent with [31], and apply adversarial attacks at test time. A learning rate of 0.0003 is used for all datasets. We perform 10 random runs for each method and report the average results. DiffSP is implemented in PyTorch with $\sigma = 2$ and $\alpha = 2$. Additional important parameter values are provided in Table D.3. More implement detailed information is available at https://anonymous.4open.science/r/DiffSP.

All the experiments were conducted on an Ubuntu 20.04 LTS operating system, utilizing an Intel Xeon Platinum 8358 CPU (2.60GHz) with 1TB DDR4 RAM. For GPU computations, an NVIDIA Tesla A100 SMX4 with 40GB of memory was used.

## E Additional Results And Analysis

### E.1 Further Analysis of Non-Isotropic Diffusion

We further analyze the core LID-Driven Non-Isotropic Diffusion Mechanism of DiffSP. In Figure E.1, we compare the sum of edge weights for clean edges during the reverse denoising process of both non-isotropic and isotropic diffusion in the IMDB-BINARY

dataset under PR-BCD, GradArgmax, and CAMA-Subgraph attacks. For clean edges, the weights are positive, whereas for adversarial edges, the weights are negative. As shown in Figure E.1, the non-isotropic diffusion process introduces more noise to adversarial edges while minimizing the perturbations on the unaffected clean structure. This results in a faster and more effective recovery of the clean structure compared to isotropic diffusion.

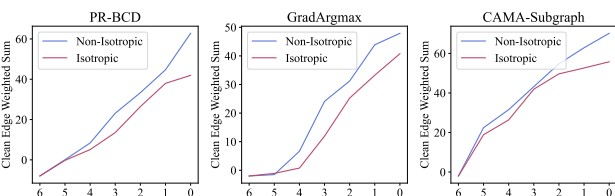

### Figure E.1: Non-Isotropic Diffusion Study

### E.2 Further Analysis of $k$ Selection

We further analyze the impact of selecting different values of $k$ in the LID-Driven Non-Isotropic Diffusion module. We adjust $k$ within 3, 4, 5, 6, 7, 8 on the IMDB-BINARY dataset under PR-BCD attacks. Figure E.2 shows the classification accuracy and the LID value ratio between adversarial and clean nodes. The red line (LID ratio = 1) indicates equal LID values for adversarial and clean nodes. All $k$ values demonstrate the ability to detect adversarial nodes. Additionally, better adversarial node detection leads to improved graph classification accuracy, as clean nodes experience less perturbation while adversarial nodes undergo more purification.

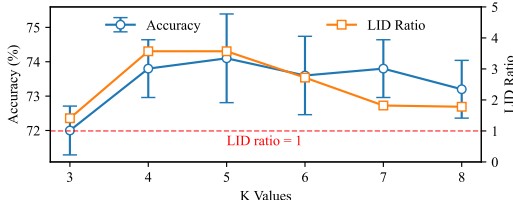

### Figure E.2: $k$ Selection Study

## F Limitations and Future Discussions

Although DiffSP enhances the robustness of graph learning against evasion attacks through prior-free structure purification, it still has certain limitations, which we aim to address in future work. Specifically: 1) In addition to structural disturbances, feature perturbations are common in real-world scenarios. In future steps, we plan to incorporate experiments on feature-based attacks and evaluate robustness in link prediction tasks under evasion attacks. 2) Estimating the adversarial degree of nodes is crucial for non-isotropic noise injection. We aim to develop a more accurate estimation method to further enhance the robustness of graph learning. 3) We also plan to optimize the time complexity of DiffSP to make it more efficient.

Furthermore, the graph entropy estimation approach proposed in this work is a promising tool. We will explore ways to enhance the properties encapsulated by graph entropy, such as designing better $\mathbf{Z}$ to capture the more local structure and feature characteristics of nodes. Additionally, we plan to utilize this graph entropy method to further investigate graph properties across diverse scenarios, facilitating more extensive research in this area.

