# OpenReview forum: "Robust Graph Learning Against Adversarial Evasion Attacks via Prior-Free Diffusion-Based Structure Purification"
_ACM.org/TheWebConf/2025/Conference — WWW 2025 Poster_

### Official Review · Reviewer_6RjP · 2024-11-30

**Novelty:** 6
**Technical Quality:** 6

**Review:**

Strengths:
- Prior-Free Design: DiffSP eliminates reliance on dataset-specific or attack-specific priors, making it adaptable across diverse scenarios.
- Novel Mechanisms: The framework combines LID-driven non-isotropic noise injection and transfer entropy-guided denoising to effectively target adversarial perturbations while preserving valuable graph information.
- Generalizability: Works well for both graph and node classification tasks, demonstrating robustness across varied applications.

Weakness:
- Lack of Baseline Comparisons: For node classification, DiffSP should be compared against IDEA [1]. For graph-level classification, it misses comparisons with HEAL [2].
- Performance on Non-Adversarial Noise: DiffSP struggles with weaker attacks or naturally noisy data, where adversarial characteristics are less distinct.
- Sensitivity to Parameter Tuning: The framework depends on careful hyperparameter selection (e.g., noise levels, diffusion time) for optimal performance, which may vary across datasets.

**Questions:**

- Could the inclusion of node features in the diffusion process further improve robustness and accuracy?
- How does the performance of DiffSP scale with extremely large graphs or real-world data with incomplete or missing edges?
- How sensitive is DiffSP to errors in adversarial degree estimation during noise injection, and what happens if clean nodes are misclassified as adversarial?
- Are there potential benefits in integrating DiffSP with other adversarial defense techniques, such as adversarial training or robust aggregation methods?

[1] Tao, Shuchang, et al. "IDEA: Invariant Defense for Graph Adversarial Robustness." Information Sciences 680 (2024): 121171.
[2] Ju, Wei, et al. "Hypergraph-enhanced Dual Semi-supervised Graph Classification." Forty-first International Conference on Machine Learning.

**Reviewer Confidence:**

4: The reviewer is certain that the evaluation is correct and very familiar with the relevant literature

**Scope:**

4: The work is relevant to the Web and to the track, and is of broad interest to the community

---

### Official Review · Reviewer_UVEh · 2024-12-02

**Novelty:** 4
**Technical Quality:** 4

**Review:**

This paper focuses on graph learning against adversarial evasion attacks. The proposed DiffSP achieves such goal by predicting the distribution of clean graphs via diffusion models. The author points out two major challenges and proposes LID-driven non-isotropic diffusion mechanism and Graph Transfer Entropy Guided Denoising Mechanism to address the aformentioned issues. Experiments on both graph and node-level tasks are carried out to show the effectiveness of the proposed method.

**Strength**
- The paper is well-writen and easy to follow.
- The proposed method is presented with strong theoretical support.
- Extensive experiments are carried out to verified the effectivenss of DiffSP.

**Weakness**
- Baselines are somewhat out-of-date. Some methods are not mentioned (e.g. [1]).
- In the experimental section, the authors mention that the model's superior performance is related to its prior-free nature. However, the causal relationship between these two factors is not well-established.

**Questions:**

- Could the author be more specific about why learning the clean distribution of graphs makes DiffSP prior-free?
- What's the difference between this paper [2] and [3]?
- Could the author provide detailed information of the computational cost and runtime of DiffSP and baseline models?

[1] Zhao, Kai, et al. "Adversarial robustness in graph neural networks: A Hamiltonian approach." In NeurIPS 2023.

[2] Zhuang, Jun, and Mohammad Al Hasan. "Robust Node Representation Learning via Graph Variational Diffusion Networks." arxiv.

[3] Kang, Mintong, Dawn Song, and Bo Li. "Diffattack: Evasion attacks against diffusion-based adversarial purification." In NeurIPS 2023.

**Reviewer Confidence:**

4: The reviewer is certain that the evaluation is correct and very familiar with the relevant literature

**Scope:**

3: The work is somewhat relevant to the Web and to the track, and is of narrow interest to a sub-community

---

### Official Review · Reviewer_SJeU · 2024-12-02

**Novelty:** 4
**Technical Quality:** 6

**Review:**

The paper presents an novel approach leveraging diffusion models to improving the robustness of GNNs against adversarial evasion attacks.

**Pros:**

1. The paper is generally well-written. The main ideas are presented in a coherent manner.
1. The authors conduct a comprehensive set of experiments across multiple graph and node classification tasks. The experiments are detailed and provide valuable insights into the performance of the proposed method.
1. The paper includes a certain level of theoretical analysis.

**Cons:**

1. While the authors provide some analysis of the computational complexity, there is no in-depth comparison of memory usage and runtime performance when using the diffusion model, especially graph. Given the computationally intensive nature of diffusion models, it would be useful to include these details to assess the efficiency of the proposed method.
2. The experiments are primarily conducted on relatively small datasets, which may not fully reflect the scalability of the proposed method. A more robust evaluation on larger-scale datasets would be valuable to demonstrate the efficiency and scalability of DiffSP.

**Questions:**

1.  How do the diffusion-based components affect the overall runtime and memory consumption?

2.  How is the model's performance on larger-scale datasets?

3.  How transferable is the learned diffusion model to unseen attack types or attacks with different characteristics?

4.  How does DiffSP generalize to graphs with very different structural properties (e.g., highly sparse vs. highly dense graphs)? Would the framework benefit from specialized strategies to handle these cases?

**Reviewer Confidence:**

3: The reviewer is confident but not certain that the evaluation is correct

**Scope:**

4: The work is relevant to the Web and to the track, and is of broad interest to the community

---

### Official Review · Reviewer_iod5 · 2024-12-02

**Novelty:** 6
**Technical Quality:** 6

**Review:**

The paper presents DiffSP, a robust framework for defending GNNs against adversarial evasion attacks. Using a prior-free diffusion-based purification process, it removes adversarial perturbations through selective noise injection and entropy-guided denoising. DiffSP shows strong robustness across diverse datasets and attack types, outperforming existing methods while ensuring semantic alignment with the original graph structure.

Pros:
1. The proposed method avoids reliance on heuristic priors, enhancing adaptability across datasets.

2. This paper introduces an innovative purification process combining non-isotropic diffusion and transfer entropy mechanisms.

3. The proposed method adopts a theoretically sound framework backed by well-defined mathematical formulations.

4. The LID-driven non-isotropic mechanism efficiently targets adversarial nodes without affecting clean ones.

Cons:
1. It struggles with weaker attacks that introduce plausible noisy edges, reducing effectiveness.

2. The graph entropy estimation process may face challenges in scalability for large datasets.

3. The reliance on multiple hyperparameters may complicate implementation and tuning, for example, the hyperparamter $$K$$.

4. It is unclear how the proposed method handles adversarial attacks that alter both structure and node features simultaneously.

**Questions:**

1. How the proposed method DiffSP handle dynamic graphs where structures evolve over time?

2. Is the method robust to simultaneous feature and structural perturbations?

3. What is the trade-off between computational cost and robustness, and are there opportunities for optimizing the process for online learning scenarios?

**Reviewer Confidence:**

4: The reviewer is certain that the evaluation is correct and very familiar with the relevant literature

**Scope:**

4: The work is relevant to the Web and to the track, and is of broad interest to the community